# The City of Muses Project: Creating a Vibrant and Sensual Metropolitan Landscape through Architecture

Antonella Contin

Department of Architecture and Urban Studies, Politecnico di Milano, 20133 Milano, MI, Italy; antonella.contin@polimi.it

**Abstract:** The Metropolitan Architecture Project aims to create an artistic metropolitan landscape, which captivates visitors. It focuses on the relationship between the form's image and the surrounding context, emphasising the structural image in architectural design. The project draws inspiration from the City of Muses Project, incorporating a symbolic mediator as a propeller, which represents the connection with the contemporary society's cultural symbols and bridges the gap between the past, present and future. The methods employed in the Metropolitan Architecture Project involve integrating artistic elements into the metropolitan landscape. This includes incorporating the symbolic mediator and designing the structural image to interact harmoniously with the surrounding environment. The project has successfully introduced a new type of built form characterised by a relational figure and a vibrant and sensual image. By embracing this approach, the architectural design actively engages with the environment and enhances the overall architectural experience. The Metropolitan Architecture Project demonstrates the significance of incorporating an artistic dimension in creating a metropolitan landscape. The project achieves a captivating and interactive architectural design by considering the dynamic relationship between the form, context and structure. This understanding of architecture contributes to a deeper comprehension of the society which constructs it, resulting in a rich and engaging architectural experience.

**Keywords:** metropolitan architecture project; art; new built form type; symbolic mediator





## 1. The Power of the Image

The Eden from *Star Wars* (Figure 1) can be envisioned as a composition of various elements. While the author of the film used the panorama of Bellagio (a), a small Italian town, as the basis for the image, it is not a mere replica of the panorama. The image's foreground prominently features the balustrade of the famous Bellagio villas in the town. With the help of technology, the author introduced new characters and manipulated the image to create the Eden from *Star Wars*.

In the second image (b), two more actors complement the original elements. The first is the Theatre of the World, a large wooden toy created by Aldo Rossi, which was placed on a boat that sailed along the Venice lagoon. This toy reacted with the city of Venice, the beautiful Piazza San Marco and the neglected Venetian area of Punta Fusina at Porto Marghera. The second addition is the two mannequins, which Giorgio De Chirico used to incorporate into his artworks, representing the Muses and the Arts. This new image aims to capture the modern picture of the metropolitan city, which emerges from a true poetic reaction. The image of modern architecture is rooted in a place and interacts with it, creating a unique and memorable reality, which stands out from the generic city (Koolhaas et al. 1997).

The goal of this new image is to capture the contemporary way of exploring the relationship between science and art, as well as the interplay between art and technique, as exemplified by the architecture. By using a science fiction cinema image, we can create a visual representation of this concept.

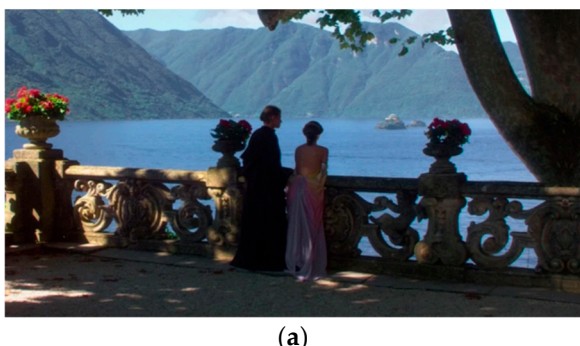
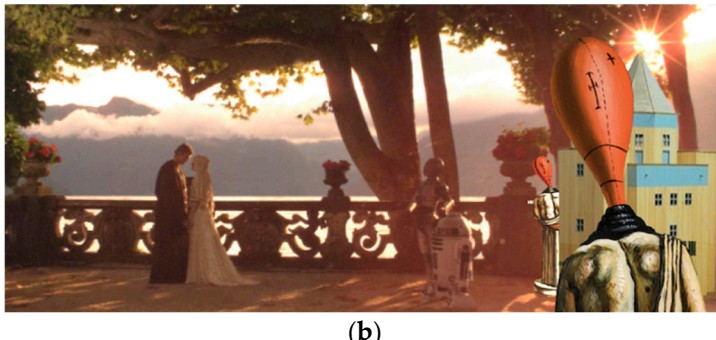

**(a)** **(b)**

**Figure 1.** (**a**) *Star Wars* Eden and the power of a science fiction image; (**b**) *Star Wars* Eden revisited.

The Theatre of the World, created by Aldo Rossi, provides a model for the type, which interacts with the Muses of De Chirico's arts, transforming the environment of Bellagio. However, we must incorporate an artistic dimension into the composition to produce a solid and emotionally engaging panorama. This will enable the visitor to become fully immersed in the scene and participate in the new reality created by the image. The aforementioned collage highlights the role of the Metropolitan Architecture Project in introducing a new concept of poetic reaction image. This image is characterised by a timeless, out-of-scale figure interacting with the surrounding context to create a panorama of a metropolitan landscape. The resulting image must be deeply expressive and affective, reminiscent of Carlo Carrà's renowned work, L'Idolo Ermafrodito (Figure 2), which depicts a colossal body in a small room.

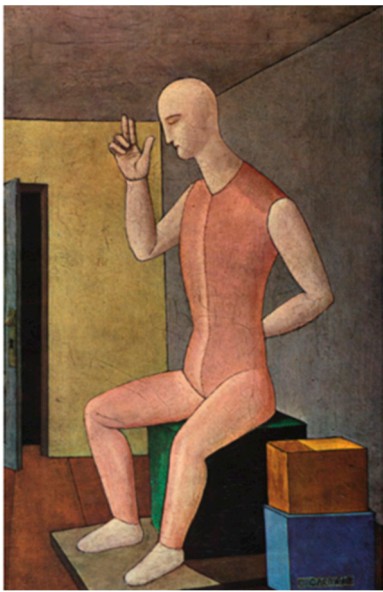

**Figure 2.** L'Idolo Ermafrodito by Carlo Carrà (1917).

This image is a metaphor for the collision between the new types of built forms—or Urban Morphotypes (d'Alfonso 2014) or Megaforms (Frampton 2021)—and the metropolitan contexts, which Rem Koolhaas calls "the Bigness". Using these poetic reaction images, the Metropolitan Architecture Project can create a new and compelling visual language for urban landscapes, which evokes a strong emotional response from the viewer while capturing the unique character and identity of the city. To achieve an engaging poetic reaction to the place, the metropolitan Urban Morphotypes must incorporate an affective reaction produced by an artistic fact. This is crucial in the second modernity architecture, as it creates a memorable and unique reality rooted in the place and interacts with it profoundly.

The metropolis of second modernity and its new architectural form types refer to the post-Second-World-War cities, which were destroyed during the war and flourished into the new capitals of second modernity in the 1950s. This shift in thinking from a destructive tool of violence to a generative artistic force led to the creation of cities as a space for artistic expression and not destruction. Metropolitan cities may become the "polis" of the 21st century, where the same laws govern all citizens and create a harmonious relationship between the city and its citizens. The new modernity must find a new way to express and communicate these values towards a new image of the city to be transmitted to the senses, where the human body perceives the limits of things and filters the consistency of reality (Rogers 1968).

Furthermore, at the Architecture Faculty of the Politecnico di Milano, E.N. Rogers, who re-founded the school after the Second World War, in the lectures he gave from 1953 to 1963 did not present a study of the type related to its ideal moment. Still, he analysed the theoretical but also practical and constructive ways of the type's concept. According to E.N. Rogers (Contin 2021), the image of architecture is just as important as its form.

E.N. Rogers departs from the traditional "typical" of the architecture of the past and instead focuses on the relationship between the immediacy of the image and new architecture. The trace of what has passed is therefore a physical trace. Still, it does not carry a syntax because if it were the object of hermeneutic and not subjective interpretation, the contribution of contemporary design as a relationship between memory/invention and context/individuality would be null and void. Ultimately, the constitution of the new architectural object is not determined by the deep structure of the ground but rather by the figures and images, which make up its footprint and elevation to the horizon, as noted by Le Corbusier (2003) and Calvino (1985). For him, the contents of the physical measure must be qualified by the presence of images.

Moreover, for E.N. Rogers, the type is not a mere mould for the form (Muratori 1967), which absorbs everything, including the architecture's image. Instead, the image of the form is an active agent, which reacts to the surrounding circumstances. This is why we speak of the form image in Milan in fields such as architecture, design and fashion. The form image is a structural image (Lynch 1960), which goes beyond the mere aesthetic commentary on the architects' plans. It is an image, which, due to its visual and spatial character, organises the footprint of newly built form types named, respectively, the New Morphotype (d'Alfonso and Samsa 2001) or the Megaform (Frampton 2021). It serves as a reference structure for a constellation of meanings, and it is also a relational component of the Metropolitan Architecture Project (Contin 2015).

The Measure and Scale Lab (MSLab) research unit at the Department of Architecture and Urban Studies of the Politecnico di Milano has developed the City of Muses Project idea, whose aim is to describe the image of an architecture subject with the aim of studying the image in order for it to be able to become a processor of the tonality of landscapes. This is our Manifesto:

> *Understanding places requires referencing the Muses, invoking words and figures, which stimulate the mind and create an image that goes beyond consumerism and connects with identity. Each image can act as a symbolic catalyst within the urban setting, functioning as a spatial device, which links the place and the event through mental perception. As such, images can reflect the fast-paced nature of contemporary life, incorporating archaeological remnants and diverse cultural symbols between the past, present and future.*

According to Kevin Lynch (1960) in *The Image of the City*, we aim to gain control over the production of the urban image by developing a visual plan for the city, which promotes the reorganisation of public mental images of the urban environment. The goal is to increase the citizen's integration with their place of residence and work. Lynch's approach emphasises a possible concrete order, which exists beneath the chaotic appearance of the urban scene and seeks to enhance it by increasing the visibility and individuality of urban forms.

The project pursued by Lynch—as well as by us—is that of urban figurability, which is formalised in a visual plan not purely limited to the aesthetic aspects but, on the contrary,

being aimed at the well-being and the psychological security of its urban users (*feeling of adequacy*) linked to their orientation skills. Figurability is the readability of visual patterns. The aim is to design a shared public mental image of the city by recognising the deep identity of selected places and portraying them, combining spatiality and arts, particularly music and images.

In this sense, the architectural project represents an essential trait of the city, as it is constitutive of its mental image.

Nevertheless, the essence of the City of the Muses project is to place the inhabitant at the centre of the space, inspired by the futurist principle of putting the spectator at the centre of the scene (Marinetti 1912). The aim is to transform the space of the new architecture into an environment, where the inhabitant's body is central, and their behaviour is directly affected by the environment. The effects are a product of the nine Muses' actions mentioned by Hesiod in *The Theogony*, which we interpreted as Erato for aesthetics and beauty, Polyhymnia for space and ground, Thalia for urban green, Melpomene for celebratory spaces, Euterpe for sound, Calliope for memory, Clio for history, Urania for visual and city skyline, and Terpsichore for movement and musicality (Hesiod n.d.).

The new architecture becomes an event that happens and involves active players in the scene who respond to the signals given by the environment. The space is responsive and continuous over time, and even a passive structure, such as an office, must ensure that users are super-active *interactors*. The project's imagery has two tracks: a syntactic and paradigmatic one, tied to the concrete data of reality, and a characterological and figurative one, which illustrates how the various actors enter the scene space, creating a strong-minded situation.

Symbolic objects as mediators reacting to a precise place define the sociological dimension and become a sign of the spirit of the times, connected with the metropolitan city scene. According to Lynch (1960), people perceive places through the senses and learn to know what encircles them. Other arts improve this conscience and give sound to the landscape, developing the meaning of a place. Overall, the aim is to create an inhabitable architectural space, which is no longer just a leisure place, such as "a party", but becomes a "trap", engaging and responding to the inhabitant's needs.

The City of Muses, or the city of art, promotes the anthropology of the inhabited metropolitan city as a form of training. Indeed, the concept of the City of Muses was created in relation to Augustine's (1995) *imagines agentes* and the archetypes, which are repeated in human biography when looking at the city of today, as conceived by Rossi (1981). Instead of focusing only on building types, we use interpretative maps based on topographical and geographical elements to detect the possibilities and local human and natural resources. We aim to "liberate" these resources from traditional urban accumulation processes—to use Marxist terminology (Harvey 2006). The word "Muse" aims to discover and invent urban facts not for narcissistic exhibition or financial gain but for the "public", which includes natives and residents, among others.

This article contends that the Metropolitan Architecture Project (Contin 2015) is not simply about creating functional places and visually appealing structures but rather about designing an emotional landscape, which can communicate a robust poetic image. The project's capabilities extend beyond the artistic values at the beginning of its conception. It can include cultural and natural aspects, encompassing the contextual reaction of the current architectural design proposals.

Recent examples of built (Weiss-Manfredi, Diller Scofidio + Renfro, Big, Snøhetta, Zaha Hadid Architects, McLaren, Excell, Nouvel, Turenscape, Martha Schwartz and many others) and unbuilt design proposals implemented within the Metropolitan Architecture Project framework have suggested positive reactions both from the cultural and natural contexts. This demonstrates how the Metropolitan Morphotype concept (d'Alfonso 2014) can define emotional landscapes, which harmonise with the environment and society.

The project's technological components are also crucial in guaranteeing the sustainable qualities and values of the same project as an applied "utopia of reality" (Rogers 1965). This helps ensure that the project's vision is not just a dream but a practical, achievable reality.

The Metropolitan Architecture Project's approach to architectural design is demonstrated in its ability to consider the dynamic relationship between the form, built and natural contexts, and structure. The project's understanding of traditional architecture composition, articulating the context's different scales, contributes to a deeper comprehension of the society, which inhabits them, resulting in a rich and engaging architectural experience. By integrating artistic elements into the metropolitan landscape, the project can create an emotional landscape, which captivates visitors and interacts harmoniously with the surrounding environment.

Moreover, does landscape urbanism result in the creation of Metropolitan Architecture Projects? This question has been explored within the discourse of landscape urbanism and the concept of Megaform, as formulated by Frampton (2021). While Megaform is derived from architecture, there may be interesting overlaps with landscape urbanism.

Some of the projects related to landscape urbanism can be illustrated according to the concept of Megaform as the starting point of a Metropolitan Architecture Project, particularly those functioning as landmark places, which are large-scale manifested expressions of their intrinsic structure, sharing a topographic, horizontal thrust of their overall profile and presenting a catalytic function for the urban pattern. Creating a new topography, which transforms the ground's surface to create a public domain and a landmark within the space endlessness of the megalopolis, is a concept applicable to projects such as West8 surfaces in Toronto Central Waterfront (2008).

Additionally, projects such as the Yokohama port terminal by FOA (2002) and the Olympic Sculpture Park by Weiss and Manfredi (2005) also demonstrate the complexity and tectonic character of the Megaform. These projects blend exterior and interior spaces, resulting in a hybrid architecture with a recognisable infrastructure landscape.

Frampton also notes the physical integration with the site as a topographic character of the Megaform, as seen in works such as the Igualada Cemetery by Carme Pinos and Enric Miralles (1992) and the Granja Escalator by La Pena y Torres (2000).

Overall, landscape urbanism represents a theoretical and cultural shift in urban design, which focuses on landscape in its ecological, technological and morpho-typological declination. The deliberate narrative of the story between humans and nature is emphasised in manipulating landscapes, which are not simply rocks, rivers and trees but architectonic surfaces subject to the new morpho-typological manipulation.

## 2. The Metropolitan Architecture Image Interacts with the Horizon, Relating to Space Time (the Sun) and the Ground

To understand the City of Muses approach, we can look to Focillon's interpretation of Jan van Eyck's famous painting, the Portrait of a Man with Red Turban (Focillon 2002; Figure 3). By changing our perspective, we can see the turban not just as a piece of clothing but also as a beautiful red velvet fabric with unique qualities, such as the beauty of its shadows and the quality of the material. Similarly, we need to approach urban and architectural design disciplines from a new perspective. Rather than returning to Rossi's idea of the archetype as the cast of the project's form, we should view the type as a relational figure formed by its physical shape and image, which interacts with the context in a localised and sensual manner (Lynch 1960). This powerful image must be memorable and desirable, allowing all metropolitan citizens, even newcomers, to produce a mental map and explore the new city-scale dimensions at a different time scale speed (Venturi et al. 1977).

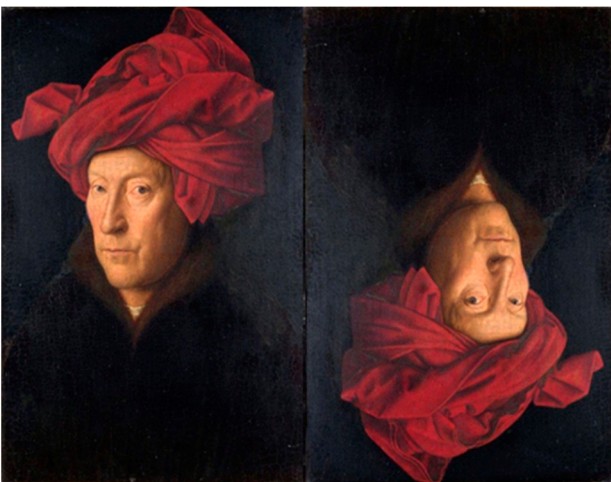

**Figure 3.** Portrait of a Man with Red Turban by Jan van Eyck (1433).

Moreover, Metropolitan Architecture Projects should not be based solely on an ideal cast; their physical volumetric consistency and footprint must react with the context like a theatre set. Their image should also react with the scene, including the horizon, sun and time. By incorporating this new approach, we can create compelling and emotionally engaging metropolitan landscapes, which capture a city's unique character and identity while making a memorable and lasting impact on its inhabitants.

According to E.N. Rogers, architecture is not simply a matter of copying the model of a type but rather involves an inventive response to contextual circumstances. In his lectures, he proposed a formula for architecture, A = f (U, B), which states that a phenomenon must balance beauty and utility over time.

This means that architecture must always find a way to satisfy the society's functional needs and aesthetic desires. Moreover, Rogers stressed that the aesthetic moment or image is not a secondary consideration, which comes after the conception of the architectural subject. Instead, it defines what architecture is as a response to its context.

To define artistic dimensions following E.N. Rogers' philosophy, it is crucial to consider the context in which a project is situated, as the image it creates is always rooted in a specific location. It is only possible to define a type by understanding its connection with the city context and the localised image it represents in response to its ground project. This idea of the ground project was introduced by Bernardo Secchi in the 1980s and reinforced the notion of the type as a reaction to contextual structures.

In addition to these concepts, Rogers introduced two key ideas through inspiring slogans: *Sperare contra spem* and *Utopia della Realtà*. The former, which is a thought from St. Augustine, refers to hope (*sperare*) for continuity in times of despair and suggests that the future can be obtained by balancing the past, present and future. This was particularly true during the Cold War era. *Contra spem* refers to the expectation of developing a context's specific values and understanding all the reasons for the society's needs. To produce a practicable project rather than a utopia, we must know all the elements, which compose the society's needs.

*Utopia della Realtà* represents a shift towards the future, as opposed to the past, in which the type is no longer defined solely by the canon or archetypes. Instead, the type recreates itself as images and footprints, which react to its surroundings. However, this raises the question of the impossibility of replicating the same figure of spaces at different times (Choay 2003, pp. 7, 8). Choay's concept of the "Figure of Space" unveils a comprehensive and fragmented chronicle of urban configurations. This narrative mirrors the profound influence of city-scale shifts, encompassing domains such as knowledge, technology, economy, politics and society.

### 3. The Building Described by the Formula with the Context

According to E.N. Rogers, context is not just a matter of syntax, as in the case of the Roman School of Muratori and Caniggia. Instead, it is a factual agent, which interacts with the architectural object in a relationship of action and reaction. The relationship between a building and its context is not based on structure but on the situation. For Rogers, the architectural phenomenon is a prototype, an object with a poetic reaction, such as the Torre Velasca in Milan, which also represents the concept of beauty in the 20th century. Rogers asserted that it is an image of discontinuity inserted into the continuity of tradition. Therefore, interventions in the city involve morphological reforms, which comment on the deep palimpsest structure of the city and its coherence of images, with a kind of transtemporal value—an iconic value:

> *"The real plan, the plan that gives a city its identity and its face, is the one that in one way or another implies the quality of individual architectural works". "The most important task facing the architects today is to apply themself to an authentic morphological renewal, in the sense of a functional invention, which is realised and produced in the formal invention, in the image".* (Rogers 1968)

In 1963, E.N. Rogers published the "Piano Intercomunale Milanese" in *Casabella-Continuità* (Rogers 1968). It was an innovative plan to organise the towns surrounding the Milan Municipality. E.N. Rogers wrote,

> *"[. . .] We must undoubtedly acknowledge the great effort of the Milanese Intermunicipal Plan [. . .]. No one thinks that architecture is the addition of forms from manuals or can be realised as a sentimental flash. Still, it would be just as absurd to expect it to result from a jumble of formulas, from specialist discussions, [. . .] that do not materialise in the spatial reality where the affirmation of forms is implicit. [. . .] In this way, thirty-eight municipalities are expressions of an instrumental process that must be evaluated for what it counts. However, it cannot give satisfaction beyond its physical measure if it is not filled with precise contents, qualified in the images. [. . .] One doubt from examining the P.I.M. is that this constant integration between the various disciplines did not have, in the final analysis, the high-level sanction given by architecture".* He also argued,

> *"The exact form glimpsed in the model, the so-called turbine, how will it be transmitted to the senses? Is it not always the human body that perceives the limits of things and filters the consistency of reality?".*

E.N. Rogers presented fresh architectural guidelines in relation to the interplay between nature, urban design and metropolitan syntax, which in turn led to new project requirements concerning technology, infrastructure, historical and geographical contexts, urban fabric and land use. This marked a departure from the static relationship between projects, historical forms and landscapes and a shift towards a new dynamic paradigm. Implementing a larger typological scale will establish an independent semantic code for architecture.

Providing feedback is essential. The renaissance perspective machine offered a means of defining the classical notion of type composition. The machine, created by Bramante, Leonardo, Brunelleschi and Alberti, allowed for exploring the relationship between art and science. Bramante's Tempietto di S.Pietro in Montorio exemplified the concept of type as a cast of an architectural model form, which Palladio dealt with in a book about Roman antiquity contained in his treatise.

However, the initial renaissance model type can be seen as a poetic response, as Bruschi (1973) designed it, to be located within a circular courtyard. This was crucial in reflecting the type's reaction as a ground-embedded tomb and the volume's interaction with the surroundings. The curved courtyard structure accommodated this contextual response, which anticipated Francesco di Giorgio's central Italian Piazza model. We should also revisit Brunelleschi's machine invention, which established the principles of perspective, which became the foundation for developing the modern camera. The

two-dimensional image produced by Brunelleschi's machine helped eliminate the initial invisibility inherent in traditional architectural drawings, as demonstrated by Palladio's section drawing, which portrays a building's reality and appearance. Let us therefore revisit the invention of Leonardo's map (Figure 4) and its ability to locate precise positions through instruments, allowing the formation of a mental map. Leonardo's maps crafted visual representations encompassing the entirety of the terrain, showcasing a profound cultural shift, which gave rise to the conception of renaissance space. This evolution laid the foundation for the creation of geographic maps, exemplified by the "Mappa della Val di Chiana", and subsequently extended to urban studies, as seen in the "Pianta di Imola", as well as typological explorations, exemplified by the "Studi Per S. Pietro". This change had a far-reaching impact on the progress of geographic, urban and architectural studies, ultimately leading to the development of typological models seamlessly integrated with their three-dimensional surroundings. These projects, ranging from the regional to the urban scale, are prime illustrations of how the renaissance revolutionised the conception of space and its subsequent evolution in geography, urban planning and architecture.

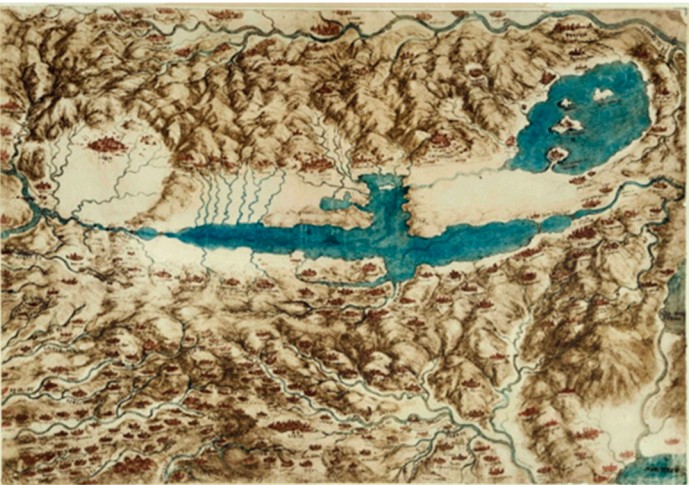

**Figure 4.** La carta della Valdichiana by Leonardo da Vinci (Windsor, RL, 12278) (1502 or 1503).

## 4. The Abandonment of Projective Architecture Design in Three Dimensions: Alberti, Le Corbusier and Gehry

Throughout the history of Western architecture, we can identify three key paradigms, which represent different eras of classicism, from the ancient, renaissance and modern periods to the contemporary era of Frank Gehry. These paradigms include the works of Vitruvius, Alberti and Le Corbusier. Each of these architects played a significant role in shaping the architectural landscape of their time, and their contributions continue to influence and inspire contemporary architects. With his bold and innovative designs, Gehry represents a departure from traditional classicism, marking a shift towards a more modern and experimental approach to architecture.

To comprehend the evolution of modern architecture, it is crucial to grasp the theoretical unease driving strategic design decisions. This tension arises from the conflicting concepts of time experienced in the present versus the metahistorical ideas embodied in a design project. This dichotomy underlies Moneo's influential choice, which heralded the second modernity's zeitgeist. Eisenman's close reading (Moneo 2004; Eisenman 2008) of architectural works and the development of exemplary prototypes over time have also contributed to the new idea of the canon and canonisation of specific design approaches. By analysing these trends, we can gain a deeper understanding of the dynamic evolution of modern architecture over time (Frampton 2007).

Bramante's Tempietto is a prominent example of renaissance architecture, which served as the central inspiration for many later buildings, including Raphael's Villa Madama, Palladio's La Rotonda and Schinkel's Altes Museum. However, the works

of Frank Gehry and James Stirling mark a departure from the traditional approach to composition, where the concept of type as a predetermined form is shattered. Le Corbusier's Ronchamp chapel project is credited with initiating this new interpretation of type in architecture. The Ronchamp is a chapel, which responds to its environment, with its form curving in response to the wind and slope of the hilltop. In the Stuttgart Gallery plan, Bramante's model is transformed into a ramp, allowing a smooth transition between the terrace and the building. Similarly, in Gehry's Peter B. Lewis Building in Cleveland, the circular form is broken and deformed, significantly departing from the traditional model.

## 5. From the Model Type to the Three Compositions

The contemporary approach to architectural composition departs from Argan's theory, which classifies the types into three categories: whole building configurations, large building elements and decorative elements. Instead, Ernesto d'Alfonso's (2014) methodology involves three stages: dispositional composition, structural composition, as well as footprint and ground connection, and distributive composition. This marks the third paradigm shift in architecture, following those initiated by Alberti and Le Corbusier. Cubism was eventually replaced by futurism, which called for a break from tradition and a fresh approach. We can see a return to E.N. Rogers' principles.

However, the contemporary approach to architectural composition requires us to move beyond Le Corbusier's classical works, which included the sequence of the five moments, the five points and the fourth plan. Le Corbusier drew inspiration from the past but used it as an image, as seen in the Attico Bestegui, where he referenced the Villa Adriana Pecile Wall and the Pantheon light for their powerful imagery, which evokes beauty and emotion, enabling the creation of poetic projects.

This brings us to the new puzzle in architecture, which Gehry's work explores. It challenges us to conceive objects without relying solely on projective drawings but by directly producing three-dimensional objects with the support of new technologies. Gehry's process, as seen in the construction of the Bilbao Museum maquette, begins with geometry but requires breaking away from traditional forms to produce something, which reacts to its situated context.

## 6. The Second Modernity Architecture Types

The new metropolitan typology approach aligns with the three hybrid types described by James Fenton, who in 1985 was able to categorise buildings based on their form and function. The three types are fabric, graft and monolith (Fenton 1985). Fabric buildings blend seamlessly into the existing urban fabric and often incorporate similar materials and design elements as the surrounding buildings. Graft buildings bridge the gap between the existing urban fabric and new development, often incorporating different materials and design elements to create a visual contrast. Monolith buildings are large, singular structures dominating the surrounding landscape and serving a specific function.

However, Fenton's definition of hybrid buildings needs constant updating following the recent advancement in new communication technologies and the architectural diversification of the present times. To interpret the Metropolitan Architecture Projects of today through the Fenton categorisation, we identified the three Fenton types in projects exemplifying the Fenton types of evolution, including the Rockefeller Center as a fabric building, the United Nations New York Headquarters as a graft building and Moneo's L'Illa in Barcelona as a monolith building (Fenton 1985).

The metropolitan paradigm (Contin and Galiulo 2020) we want to foster is a conceptual framework, which identifies and develops central areas within a city characterised by unique features. Hybrid architectural entities, such as Urban Morphotypes (d'Alfonso 2014), Heterotopias (Shane 2005) and Megaforms (Frampton 2021), represent a modern approach to hybrid buildings, which interact with the landscape and contribute to creating distinctive configurations, forming mental maps at the metropolitan scale and shaping the emergence of a "body space" (Shane 2005) within the metropolitan fabric.

The perception of open space shifts from the background to the figure in constructing a mental map, reimagining the void or "space in between" at the metropolitan scale. The formal paradigm of this "space in between" allows for recognising conventional open space forms and creating a gradation of public, semi-public, common and private spaces.

Hybrid architecture typology has gained significance in meeting urban actors' diverse choices and needs. Hybrid buildings accommodate multiple choices, promote flexibility and respond to the needs of the contemporary city. They represent change and innovation, combining different functions and reflecting the concept of heterotopia as a recombinant architecture (Shane 2005).

The characteristics of hybrid buildings extend beyond their functions, involving complex functioning modes and internal dynamics. Recent advancements in hybrid architecture explore new formal solutions, moving away from grid dominance. Hybrid buildings interact with the surrounding context, combining public and private spheres, accommodating lifestyles and promoting sociability.

To define a metropolitan quality space, a hybrid building must be situated in a specific context with a strategic action design programme. It requires a recombinant three-dimensional local matrix to stabilise or destabilise urban dynamics. The Metro-Matrix Metropolitan Acupuncture Chart (Ortiz 2014) guides the positioning of hybrid buildings as ordering agents at the metropolitan region scale.

The new hybrid season of the metropolitan city redefines space quality by defining permeable boundaries of heterotopic enclaves to reflect the city and provide access to diverse urban actors. Categories such as reflexivity, public sociability, formality, processualism, interscalarity, density, intersomaticity, measurability and combinability, aid in evaluating the quality of Metropolitan Morphotype spaces.

As a matter of the present times, the complexity of these new forms increases due to the linkages between urban–rural and urban–natural patterns, which share characteristics with Frampton's concept of the Megaform, which can inflect the existing urban landscape due to its stable topographical features. Ultimately, this new public realm relates to the metropolis, achieves a unique aesthetic and creates a new sign for local and metropolitan scales.

In his collaborative work with Iturbe *Lateness* (Eisenman and Iturbe 2020), Eisenman raises a cautionary flag against the possible pitfalls associated with a novel architectural approach, which can be linked to the eco-morphology production introduced by Bjarke (2013). He emphasises a significant transition from perceiving architecture as a complex and layered masterpiece, replete with profound symbolism, to likening it to a more simplistic entity like "a potato". Eisenman contends that architecture, functioning as a means of notation, conveys messages and ideas through symbols and visual representations. In this light, he argues for recognising architecture as an artistic pursuit rather than a mere craft.

Architecture is a language, which communicates the relationship between space and time and how they can be synchronous in Soja's "third space". With the advent of digital technology, the relationship between space and time needs to be reimagined, as digital forms can appear timeless, and space tends to flatten out in time.

To understand the new forms of built environments, we must take into account the changing dimensions of cities. The Barcelona block designed by Cerda, which had dimensions of $110 \times 110$ m, has given way to larger urban sectors in the Barcelona Plan Macia, ranging from $400 \times 400$ to $400 \times 1200$ m, and even larger megablocks of around $1 \text{ km} \times 1 \text{ km}$, as described by Professor Shane. These new dimensions result in new relationships between the buildings, infrastructure and neighbourhoods produced at infrastructural nodes. We also need to consider new building–street relations, where complex buildings manage city dimensions through multiple-scale invariants, creating a different connection with the ground. This includes a new relationship between the buildings and roads, moving away from Cullen's city scape vision to the directional space legend conceived by Venturi. In addition, contemporary flows require a leap of scales and places to create a sense of security and confidence in their internal landscape, as demonstrated by L. Kahn's Philadelphia civic forum and traffic study.

The New Morphotype involves new relations between buildings and the ground, which define multiple pockets of spaces characterising new function patterns and the building as a machine layer. Heterotopias (Shane 2005) are the new hyperlocal centralities, which offer different types of building forms and land use places to find the permanence of scale values and different times. They are living and connecting spaces, where the realm of the contemporary public exists. To conceptualise these new built environments, we must consider the physical and environmental characteristics of the heterotopias. Heterotopias are also spaces of flow, which require marking images on the ground and managing the relationship between different levels of the infrastructure to achieve an understandable and efficient metropolitan scale. They are consolidated public realms in the form of meeting places, which are the products of new intelligence, introducing the complexity and dimension of the landscape on a small scale.

## 7. The Project Is a Speaking Map

An exclusively technical interpretation needs to capture the true essence of a Metropolitan Architecture Project as a language-based map. Such project is more than a mere collection of structures, signs and symbols; it is a living entity, which tells the story of the life pulsating within it. The signs on the map function much like words in a language, but they are more than that. They are an expression of the collective dream of understanding and sharing our world, society and behaviours.

Our Metropolitan Architecture Project seeks to create a map, which transcends the technical constraints of military maps. Rather than following a linear structured format, it will be composed like music, with a cyclic movement, wave, breath and alternation. This immersive experience map will be a place where the beholders can find themselves "inside" and move in the rhythm and direction of their choosing. The project is envisioned as the inaugural embodiment of the rite of baptism, where the beholder is immersed in the name. In the same way, the map names the places, giving them the power to connect people with their environment.

This project will map life, the earth, the afterlife and the practices the project allows, placing the intersection of humans and the environment at the centre. It will be a map, which captures the richness of life and connects us with the world around us. Rather than being a mere manufacturer of quantitative requirements, the project as a map will process the environmental qualities, adding depth and richness to the lived experience.

## 8. Recognisable Image and Mental Map at the Metropolitan Scale

Ernesto d'Alfonso (2014) argues that Le Corbusier's encounter with the megalith on the beach of Brittany was a turning point, which marked a departure from Alberti's renaissance era perspective. The megalith's presence in its unique and irremovable tectonic and tactile environment opened up new possibilities for architecture, and its horizon line offered a glimpse of the remote setting beyond it. The belief is that this discovery invalidated the building as a signifier. A new dynamic architecture vision emerged, which emphasised the Promenade Architecturale, or the walk through the built environment, using all the senses to "perceive" the world by inhabiting it. Reeve and Simmonds (2000) describe this as the "public space as a movie set", and Kant calls it the "architecture of thought", where the act of walking through space and time is the first modality of experiencing a place. At a metropolitan scale, Shane (2005) argues that New Urban Morphotypes, such as Heterotopias, can be transformed into sites, which create a sense of belonging for all metropolitan citizens. These sites should be explored and recognised as places on the city maps worldwide, characterised by "memorable and sensual images" (Lynch 1960). The use of drawing is still essential but not as a sole signifier; instead, the building must be founded and rooted in the ground and react with the city skyline.

### 9. The Metropolitan Images: "Deposit of Imagery"

Cinema is a 21st-century art form, which allows us to conceptualise how to construct the anthropological image of the metropolis in the second era of modernity. There are two approaches to producing a constructive image of metropolitan urban design. The first is the "deep structure" type, which produces a collective memory consisting of fundamental principles related to structural features, such as building–street relationships, collective transport systems, pedestrian access and the ability of the built environment to adapt over time (Figure 5a). The second is the superstructural type of the deep structure (Figure 5b), which is particularly relevant today, as each new era or urban generation brings new values regarding the visual character and new demands regarding the function of the built environment. This is essential to be able to order the space through the rhythm of images (Le Corbusier 2003; Bachelard 1975) and to understand the incremental metropolitan growth and the time scale speed (Venturi et al. 1977).

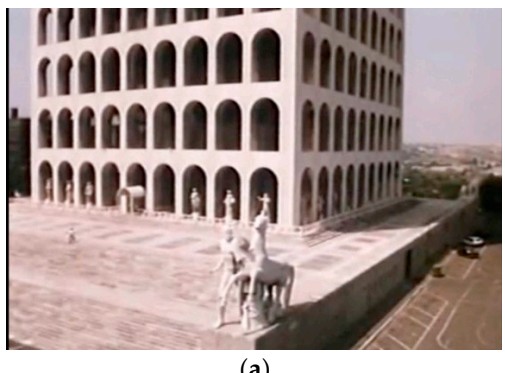

(**a**)

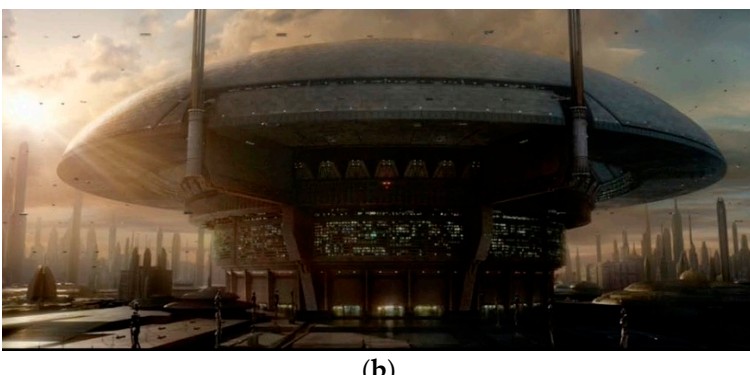

(**b**)

**Figure 5.** (**a**) *The Belly of an Architect* by P. Greenaway (1987). The EUR district in Rome, Italy. Palazzo della Civita' Italiana by Guerrini, La Padula, Romano (Rome, EUR, 1938); (**b**) *Star Wars III—Revenge of the Sith* by George Lucas (2005).

The interplay between original communication and information in today's interaction and flow space is shaped by the increasing complexity and recognisability of the local within the global context, which demands a return to simplicity and immediacy in design (Reeve and Simmonds 2000). For designers of Metropolitan Architecture Projects, the primary concern is no longer the image itself but how the figure of identity is imprinted on the mental map. The dominant trend now prefers easily organised images, which everyone can quickly understand and systematise. Additionally, places within the space of flows/communication serve as containers for internal urban landscapes, which offer attractive backdrops for citizens who experience them in a distracted temporality, drawn in by the extraordinary iconic typification disseminated through the media. As a result, there is a physical identification of places through images, thanks to an expanded sensory perception.

To avoid the risk of reducing architecture "from the jewels box into the potato" (Eisenman and Iturbe 2020), we need to view architecture as an art rather than just a craft and as a notational tool, which communicates through images and symbols with those who build it. Eisenman argues that architecture is a language, which communicates space and time, demonstrating how they can be synchronous. Therefore, we need to re-examine the relationship between space and time, especially in the digital age, where digital forms seem timeless, and space flattens out in time. Moreover, it is crucial to develop genetic systems in Metropolitan Architecture Projects, which define a recognisable image for a mental map, even at a metropolitan scale. The tall buildings dominating the metropolitan landscape are more than just functional structures; they are also genetic systems, which can shape the face of the territory and the city itself. By studying these buildings as territorial land use units and landscape types within the metropolitan paradigm, we can understand how they contribute to a recognisable image and a possible mental map of the metropolitan

city. Instead of focusing solely on their function, we can view these buildings as nuances of interior landscapes.

However, the issue arises when considering how these buildings are located and how they instruct spatial relations within the incommensurable dimensions of an urban situation. Drawing upon the disciplinary concept of "type", we can critically analyse the complexity of contemporary urban landscapes in metropolitan areas characterised by diverse forms, functions and meanings. The range of sizes and times found in these contexts—from very large and distant to minimal landscapes and from fast/slow to acceleration–pause–deceleration—reveals a multitude of rhythms, layers and narratives, which interweave the private and public spheres of metropolitan citizens' lives.

We can find some seeds of contemporary city images by looking carefully at the context image construction of some famous films. In *La Notte* by Michelangelo Antonioni (1961), the Gio Ponti skyscraper establishes the image by reacting with the old and new city dimensions able to constitute a mental map hinge point at the metropolitan scale (Figure 6a).

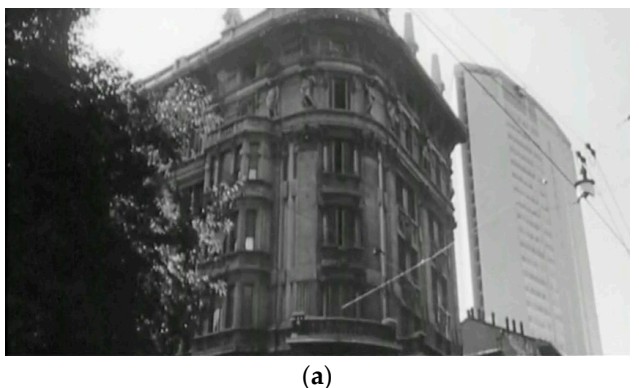
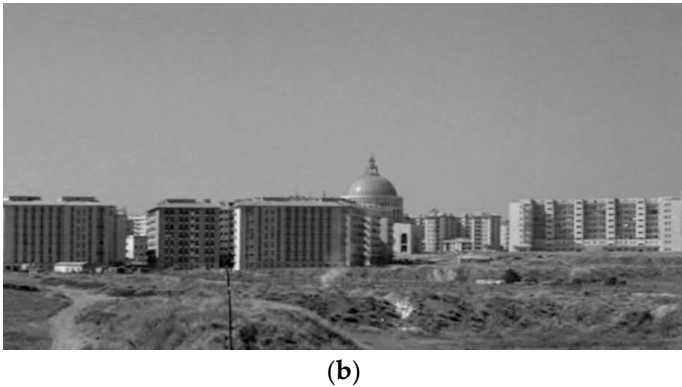

(**a**)                                                                                 (**b**)

**Figure 6.** (**a**) *La notte* by M. Antonioni (1961); (**b**) *Mamma Roma* by P. P. Pasolini (1962).

Urban Morphotypes have become the new emblematic feature of territorial expansion. As depicted in Pier Paolo Pasolini's *Mamma Roma* (1962) (Figure 6b), they serve to illustrate and signify the organisms, which shape the growth of a city and have spatial, economic and social ramifications for the surrounding region. However, their significance is often limited and distorted by a purely functionalist and quantitative approach, which fails to appreciate their added value as landmarks, icons and inner landscapes. Furthermore, there is a lack of understanding with regard to the relationship between the character of these icons, landmarks and interior landscapes and the space of flows and communication, which define the present-day public space of a second modernity metropolis. The interplay between the scale of networks and the speed of communication—and how this affects the mental maps of citizens—has not been adequately explored.

The question of urban quality is no longer solely related to the complexity of functions and building plans but also intimately tied with the identity and meaning of the image portrayed (as shown in films such as F. Lang's *Metropolis* in 1927 and D. Cannon's *Judge Dredd* in 1995 Figure 7a,b). The changing actors in the urbanisation process, along with the shift in spatial and temporal measures, have resulted in a transformation of the city's structure, including its relationships with citizens and time.

The way people experience and mark places and territories has changed, as seen in movies such as *Blade Runner* by Ridley Scott in 1982 (Figure 8a). The expansion of technologies and the meta-city image have opened up new horizons and choices for people, not only in terms of physical destinations but also identity and performance. The need to design a monumental space representing the city's identity and functions as a local space brand and a sign of mental maps linked to the metropolitan city's scale is crucial, as demonstrated in D. Cannon's *Judge Dredd* in 1995 (Figure 8b).

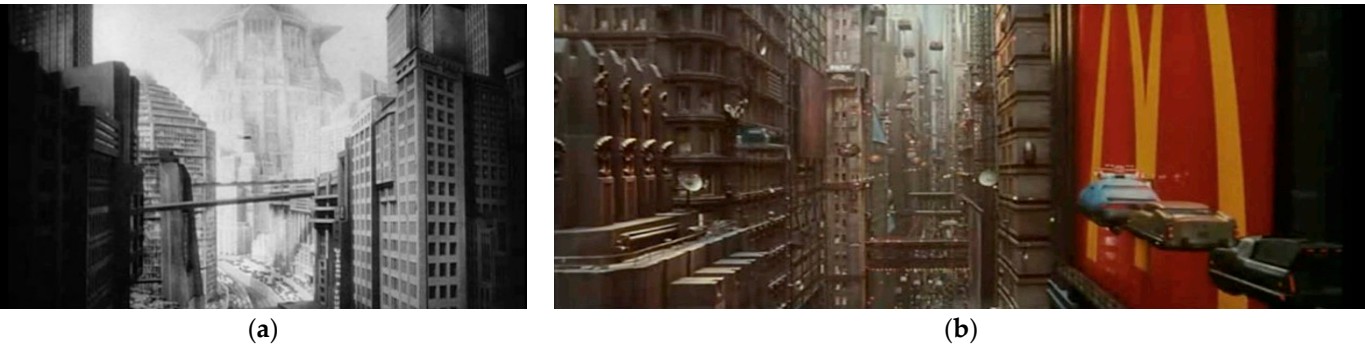

**Figure 7.** (**a**) *Metropolis* by F. Lang (1927); (**b**) *Judge Dredd* by Danny Cannon (1995).

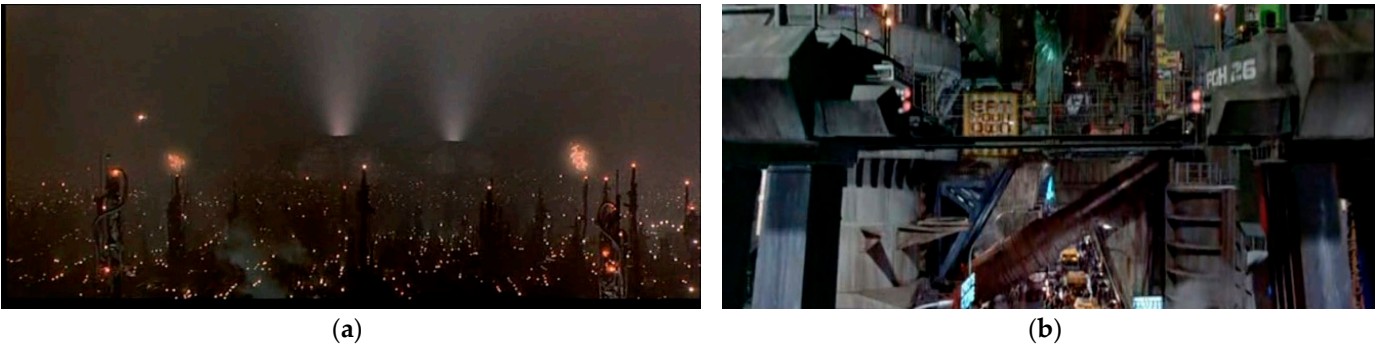

**Figure 8.** (**a**) *Blade Runner* by Ridley Scott (1982); (**b**) *Judge Dredd* by Danny Cannon (1995).

In addition, there are essential questions to consider when welcoming new citizens into the city. How can the city space be adapted to meet their expectations? How can it be designed to be desirable from a distance and yet inviting and fully accessible to local citizens up close? (*Lost in Translation* by S. Coppola (2003)) (Figure 9a).

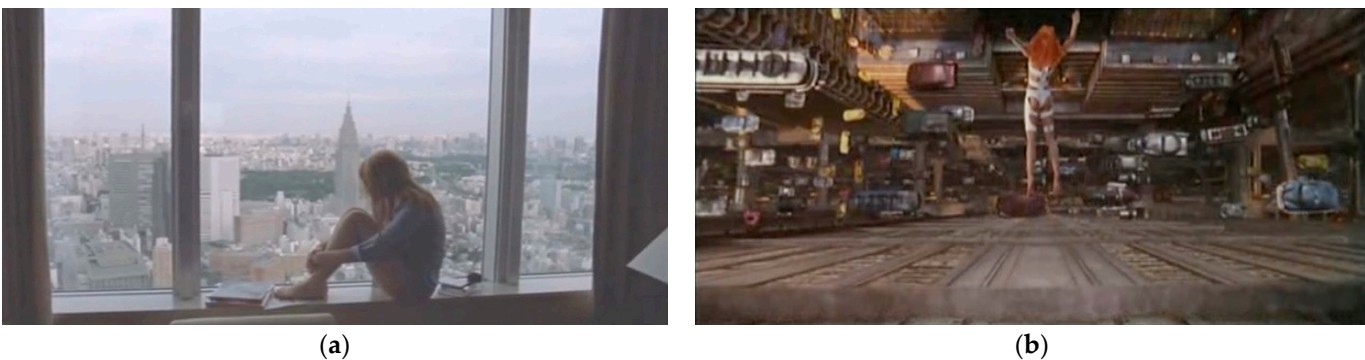

**Figure 9.** (**a**) *Lost in Translation* by S. Coppola (2003); (**b**) *Le Cinquième Élément* by Luc Besson (1997).

The goal is to understand how skyscrapers can be incorporated into the city's mental map to create a shared public image, which promotes urban users' psychological well-being and mental security. This is closely tied with orientation skills (*Le Cinquième Élément* by Luc Besson (1997)) (Figure 9b). The "footprint reaction" of the tower is an essential element, which organises an urban section (Figure 10). The building's response to the skyline creates a "landmark" or a visual connection, contributing to the visibility of the total landscape. It becomes a signal of popular communication—a value, which structures the urban image due to its visual readability and role in shaping visual matrices (*Manhattan* by W. Allen (1979)) (Figure 11).

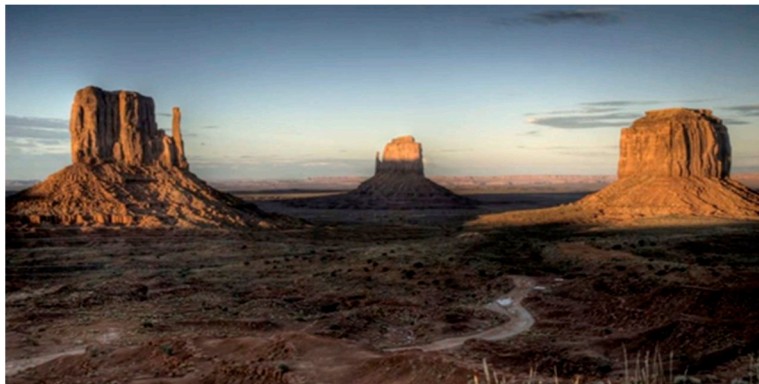

**Figure 10.** *Paris, Texas* by W. Wenders (1984).

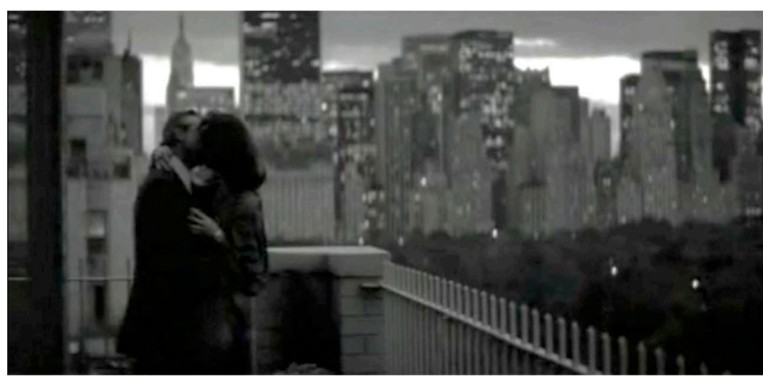

**Figure 11.** *Manhattan* by W. Allen (1979).

Ultimately, the Metropolitan Architecture Project must also focus on creating an interior landscape, which intensifies the time of metropolitan citizens (*Le Vertige* by R. Mallet-Stevens (1926); *Total Recall* by Paul Verhoeven (1990)) (Figure 12a,b). Architects should take inspiration from R. Piano's *New York Times* Building (2007) (Figure 13a) and Mies van der Rohe's and Philip Johnson's Seagram Building (1958) (Figure 13b) to create new typologies of interior landscapes rather than just focusing on the building's specific function.

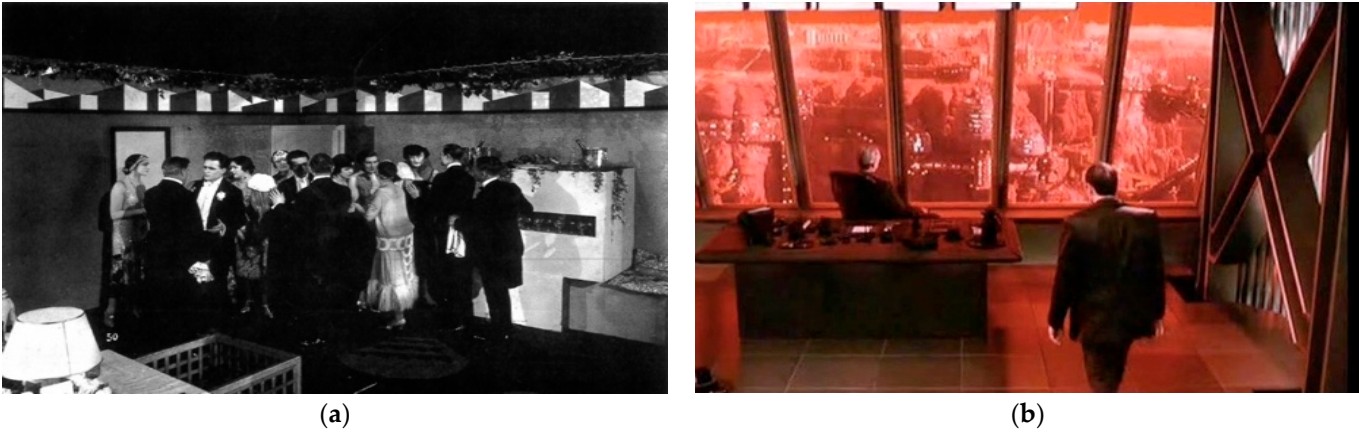

(**a**)　　　　　　　　　　　　　　　　　(**b**)

**Figure 12.** (**a**) *Le Vertige* by R. Mallet-Stevens (1926); (**b**) *Total Recall* by Paul Verhoeven (1990).

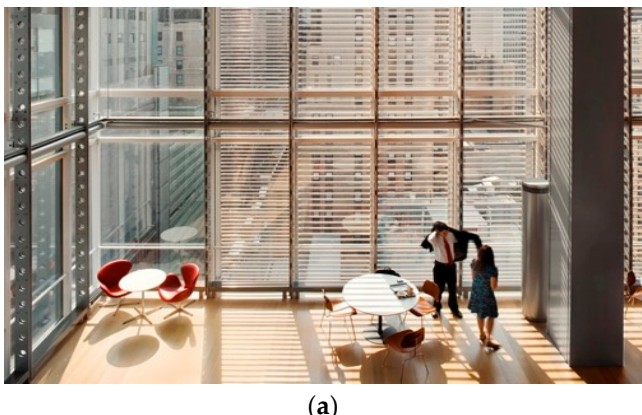
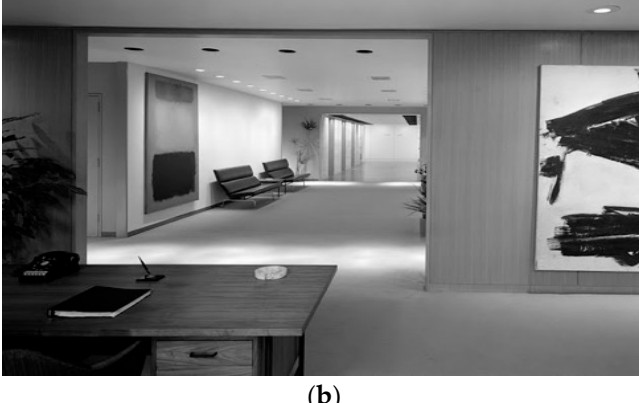

(**a**)　　　　　　　　　　　　　　　　　　　　　(**b**)

**Figure 13.** (**a**) *The New York Times* Building by R. Piano (2007); (**b**) Seagram Building by Mies van der Rohe and Philip Johnson (1958).

## 10. Conclusions

Although metropolitan cities are now many, no metropolitan area has a solid visual character or a prominent structure. Even famous cities suffer from shapelessness on their periphery (Lynch 1960). Whether a consistently figural metropolis is possible or appreciated if it existed, there are no examples, assumptions or past projections used to discuss this possibility. Nevertheless, humans have continuously expanded the scope of their perception in the past when faced with new challenges, and there is no motivation to stop this trend.

The design of metropolitan architecture provides a framework for the metropolitan urban image, and new space sequences illustrate how such systemic organisation can be achieved on a new scale. The environmental image comprises three components—identity, structure and meaning—which are always present together. For the image to be compelling, it must have a structural quality, where the parts are arranged and interrelated, thus making the object identifiable, related to the observer and with meaning for the observer, be it practical or emotional. According to Lynch, figurability is the quality, which gives a physical object a high probability of evoking a vividly identified, powerfully structured and highly functional environmental image in any observer. A highly figurable city would be distinct and remarkable, and it would invite the attention and participation of individuals, simplifying and deepening their sensory experience.

Metropolitan Architecture Projects can be considered the landmarks and footprints, which can organise the ground floor of the city and form images, which can be distinguished by their structural quality in a given context. It is a ground project. According to Secchi (1986), a ground project involves considering the surface characteristics where buildings will be constructed as part of a Metropolitan Architecture Project. A ground project must also consider the history and geography of a place, with traces of these factors being marked on the map. Diana Balmori's book *Groundwork: Between Landscape and Architecture* (Balmori 2011) explores the interrelationship between landscape and architecture, including the groundscape, as a key element for creating meaningful and functional environments; it is the ground plan's overall visual and spatial character in a particular landscape. It serves as the foundation of a landscape upon which other design elements are built, providing the context for human activities and experiences in that space.

After interpreting the construction of iconic places in cinematic art, I return to the concept of the Urban Morphotype. It is the foundation for a global relationship in the local context and a remarkable place on a mental map, which provides a link to maps at different scales. The image of the morphotype is rooted in the background, as it is obtained from it, allowing it to be contemplated from afar and from an indefinite distance.

The image has primacy for mental mapping, referring to an accurate local map (*mappa al vero*) and a single site, which the building occupies with its footprint. Landmark is the term used to describe the morphotype, contemplating its nature as an inscription in the soil

of that structural image or form image rooted into a place but responsive to that horizon and contemplated in photography or the mind as a mark of new mapping. Soil tectonics disposes itself to the landscape as a landmark of Megaform (or Groundscape), which disposes to the contemporary needs of the programme its interior as a container, but a differentiated container, which can give the form and place to necessary and desired spaces.

In conclusion, the image described in the text represents the construction of the modern city through a poetic reaction object. It is related to the architectural concepts introduced by E.N. Rogers, particularly the ideas of "Sperare contra spem" (to hope versus hope) and "Utopia della Realtà" (utopia of reality). These concepts challenge the traditional archetypes and propose a new approach to architectural design. The image portrays a shift away from replicating past archetypes and instead focuses on creating a new type, which interacts with the specific context of the city. The image of architecture becomes dominant and is not just about visibility but also about understanding the structure and meaning behind it. The image is not static but interactive with the horizon, time and space.

The "poetic reaction object" concept supports the construction of New Built Form Types, emphasising the creative and innovative aspects of modern architectural design. Overall, the image showcases a departure from traditional architectural approaches and celebrates the exploration of new forms influenced by the context and needs of the modern city. It encapsulates the vision of E.N. Rogers and the Milan Architecture School, which values the relationship between architecture and its surroundings, creating meaningful and interactive structures rooted in the present and offering hope for the future.

Nevertheless, the Metropolitan Architecture Project aims to create a metropolitan landscape, incorporating an artistic dimension to captivate visitors. The form's image is an active agent, which interacts with the surrounding context, and the structural image is a relational component of the architectural design. The City of Muses Project introduced the concept of a symbolic propeller, which connects with the fast-paced nature of the contemporary society's various cultural symbols, forming objects, which bridge the gap between the past, present and future. By embracing the City of Muses approach, the new type of built form is perceived as a relational figure defined by its form. It interacts with the environment through a vibrant and sensual image essential to understanding the work of architecture and the society, which constructed it.

Finally, metropolitan cartography plays a significant role in this process, as it serves as a methodological and technological tool to construct an image of the narrative architecture of the metropolitan territory. The type is seen as a relational figure, where its form interacts with the local context, resulting in a vivid and sensual image, which is both desirable and memorable.

**Funding:** This research received no external funding.

**Conflicts of Interest:** The author declares no conflict of interest.

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
