# Peer review of "The City of Muses Project: Creating a Vibrant and Sensual Metropolitan Landscape through Architecture"

_arts, 1938_

Round 1
Reviewer 1 Report
- I would suggest expanding on the definition of "Metropolitan Morphotype" and it's capabilities of including artistic values art the beginning of this essay, also introducing some recent examples of built unbuilt design proposals.
- The contextual reaction of the current "Metropolitan Architecture Project" concerning both cultural and natural aspects, can refer to the capability to define emotional landscapes that give a "strong poetic image" also to the technological components that are crucial in guaranteeing the sustainable qualities and values of the same project as an applied "utopia of reality"
- In regard to the "Ground Project" it could be perhaps useful for the author to refer also to the B. Zevi's book "Paesaggistica e Linguaggio Grado Zero dell'Architettura" and offer a comment on that in relation to the topic analysed in this essay.
- In regard to the concept of "Sperare contra spem," it could be perhaps useful to refer also to Massimo Cacciari's eulogy for Manfredo Tafuri titled "Quid Tum" which offers an interesting interpretation of it.
- In regard to the "New Morphotype" as catalyst of building and ground dimension, it could be interesting to include also some reference to Kennet Frampton's book "Megaform as Urban Landscape."
- The figure 5 at page 9 shouldn't be the Libera's Palazzo Dei Congress, but the 1938 Palazzo della Civita' Italiana by Guerrini, La Padula, Romano, still in Rome EUR.
- In regard to the concepts elaborated in page 10 and 11: a "groundscape" can be as strong as a "skyscrape" and perhaps today even more effective and needed in defining mental maps and orientation systems with stronger experiential qualities and not only visual. The visual, structural, and functional qualities and processes initiated by newly built/rebuilt/reused landscapes are crucial in the definition of a "third generation" of modern architecture which needs to be more resilient, sustainable, and inclusive of the first two ones!
I would suggest some small editing of the grammar (es. use of the genitive) and to shorten some of the sentences simplifying the syntax.
Author Response
1) I expanded the concept of metropolitan architecture project with regard to both cultural and natural aspects, within the urban morphotype issue.
2) I agree that while Zevi is an author mentioned in my biography, his ideas can contribute to the discussion, especially in relation to the Milanese environment, which represents both the structural and aesthetic values embedded in the concept of the metropolitan morphotype. Nevertheless, I caught the ground project concept from Secchi and Gregotti. I guess the Milanese environment well represents it, probably it is less connected to the art and manipulation of the soils, and that is true.
3) I’d prefer to quote the original author and thought. However, Cacciari’s eulogy offers an interesting interpretation of this concept.
4) Although Frampton has already been mentioned in the essay, reiterating his perspective reinforces the argument, I agree.
5) Oops! Ok, thanks.
6) I guess the ground project concept was quite exhaustive, however, I added a paragraph on the groundscape concept; Thanks.
Note. In red in the manuscript.

Reviewer 2 Report
This is an important paper that badly needs editing and the addition of a glossary of terms used. It should be accepted with amendments in my opinion.
This paper is important because it argues that a "new modernity" is emerging in Urban Design from 3 previous visual paradigms/regimes. It presents a case study of this emergence in the Urban Design eduction offered in the Polytechnic of Milan Urban Design laboratory. The paper traces this evolution of urban design ideas and pedagogy from the Post-war years of Prof. Earnesto Rogers teaching, to Prof. Aldo Rossi in the 1970's-80's and then on to the more contemporary theories of Prof. Ernesto D'Alphonso. The paper places this theoretical evolution into its local and international context. and then details thesis of a "new modernity" in terms of contemporary theory (especially Eisenman and Ghery etc).
The editorial problem is that the paper begins with a long detailed analysis of a"new modernity", utopic iconographic image and methods of composition based on the cinema creating an incomprehensible matrix of new terms and references. The authors then turn to the genealogical history and the historic pedagogical figures of the Milan Polytechnic as predecessors of the "new modernity". This outline of this overall trajectory could be briefly mentioned at the beginning with a simple example given, such as the Carra figure#2 , as an initial marker of the power of image and scale and context, bigness and room etc. This could lead in to the perspective discussion and further genealogy of teaching culmination in the break for the "new modernity". The detailed analysis of the new imaginary would then make much more sense and form a far better conclusion than the present formulation, ending in the generic interior spaces of Mies and Philip Johnson's New York skyscraper towers.
The celebration of the "new modernity" imaginary and collage capacities would then make more sense and should be followed by a Glossary of new terms , key words, such as "Eden", "Muses", "the Arts", "form image""structural image", "metropolitan landscape" etc, extending later into "composition", "turbine", "deep structure type", "meta historical", "model type","factual agent", "city identity","living or speaking map"etc.
In terms of the references used there are a few details that need correction. I would add these comments...
1. No references are given for the "Muse", I would suggest the Warburg Institute and specifically the essay about Gertrud Bing Warburg's "Muse"..There is a wonderful youtube video"Gertrud Bing and Any Warburg; Sharing the Denkraum" that could be consulted from the recent German research on Warburg. It also raises important contemporary gender issues in reading the "Muse" and the "Arts" that could strengthen the basic argument of the paper.
2. Choay is referred to twice, once I believe to her thesis on the Rule and Method, a second time to D'Alphonso's translation of another text. This needs to be clarified. The earlier work is very relevant in terms of a bi-polarity it establishes, between the arts and technology, between fixed models versus, flexible recombinatory systems, and then hybridizes using heterotopias.
3. Fenton's Hybrid Buildings is mentioned and European examples are given rather than the original American ones that lead to the skyscraper. I would argue the lines around line 260 should be checked carefully and this could also aid the transition to the use of skyscrapers later as "symbolic intermediaries" in the" identity" of cities or "metropolitan landscapes". Fenton used the Chicago Auditorium Building as the Fabric model including a tower. The Moneo project, now expanded into a Westfield megamall does make a street wall but hardly a block of the traditional city, many critics liked this hybridity that opened out to car based periphery in theory, despite its central location.I would suggest this as a Fabric scheme, horizontal and street based. Fenton used a Frank Lloyd Wright hybrid office/residential tower where each floor was hybrid around a core as his Graft model. The Rockefeller Tower is not of this same "type". The elements are clearly separated, not grafted together along their edges. Then the Moneo street facade might be seen as a monolith, but Fenton used the SOM 875 N Michigan Tower that stacked apartments above offices and a car park and mall, with a restaurant and tv station on the roof as his mono form example.
The problem with the English is the use of Italian technical terms developed in the academic community to designate various attributes of cities and urban environments. These terms in italian have a cultural and poetic, artistic dimension that does not immediately translate into English causing jarring jmpss in sense on occasion. This is why I suggest a Glossary at these dog the article, something my edits suggested at the end of my own second book.
Author Response
1) In my lecture on the City of Muses, I frequently reference Warburg's "Muse" as a representative of the muse Mnemosyne. However, referring to the Muses my cultural background's connection is to Greek and Latin literature. The interpretation of Hesiod's nine Muses has provided a metaphorical framework for potentially defining the city from an 'aesthetic' perspective. I find the comment on the gender issue intriguing. Although I am unable to expand on it at the moment, I am eager to delve deeper and explore its significance.
2) I did not quote the Rule and the Model, but only Espacemants in my discussion. In my opinion, Choay's first book is significant in the debate surrounding the Latin approach, which focuses on the rule of the city form and the production of treaties. Conversely, it sheds light on the Anglo-Saxon perspective, which is more concerned with the relationship between the social model shaping of the city. Furthermore, it explores how utopias serve as texts that can illuminate this connection. However, I understand that this particular argument may not be highly relevant to the current discussion.
3) In light of new communication technologies and the diversification of architectural practices today, Fenton's definition of hybrid buildings requires an update. To interpret contemporary metropolitan architecture projects using the Fenton categorization, we have identified three types that exemplify the evolution of Fenton's concepts. These include the Rockefeller Center as a Fabric building, the United Nations New York Headquarters as a Graft building, and Moneo's L'Illa in Barcelona as a Monolith building. To further advance the Metropolitan Paradigm that we aim to promote, we propose a conceptual framework that identifies and develops central areas within a city characterized by unique features. This framework incorporates hybrid architectural entities such as Urban Morphotypes, Heterotopias, and Mega Forms. These entities represent a contemporary approach to hybrid buildings, which engage with the surrounding landscape and contribute to the creation of distinct configurations. They play a vital role in forming mental maps at the metropolitan scale and shaping the emergence of a "body-space" within the fabric of the metropolis.
4) Regarding the Glossary, I tried to explain the keywords in the text.
